# Targeting Hypoxia: Revival of Old Remedies

**DOI:** 10.3390/biom11111604

**Published:** 2021-10-29

**Authors:** Nuria Vilaplana-Lopera, Maxym Besh, Eui Jung Moon

**Affiliations:** Department of Oncology, MRC Oxford Institute for Radiation Oncology, University of Oxford, Headington OX3 7DQ, UK; nuria.vilaplanalopera@oncology.ox.ac.uk (N.V.-L.); maxym.besh@oncology.ox.ac.uk (M.B.)

**Keywords:** hypoxia, cancer, tirapazamine, hyperthermia, carbogen breathing, tumour metabolism

## Abstract

Tumour hypoxia is significantly correlated with patient survival and treatment outcomes. At the molecular level, hypoxia is a major driving factor for tumour progression and aggressiveness. Despite the accumulative scientific and clinical efforts to target hypoxia, there is still a need to find specific treatments for tumour hypoxia. In this review, we discuss a variety of approaches to alter the low oxygen tumour microenvironment or hypoxia pathways including carbogen breathing, hyperthermia, hypoxia-activated prodrugs, tumour metabolism and hypoxia-inducible factor (HIF) inhibitors. The recent advances in technology and biological understanding reveal the importance of revisiting old therapeutic regimens and repurposing their uses clinically.

## 1. Introduction

The presence of low oxygen concentrations, known as hypoxia, is a prevalent characteristic of the microenvironment of solid tumours [1]. Tumour hypoxia causes aggressive tumour progression and treatment resistance, which leads to poor patient survival. Hypoxia arises from an imbalance between oxygen consumption and delivery, which is caused by rapid tumour cell proliferation and an inefficient blood supply carrying oxygen and nutrients [2]. In radiobiology, tumour hypoxia, which is defined as an oxygen tension below 10 mmHg, is significantly correlated with radiation resistance due to the role of oxygen in the ”fixation” of DNA damage. In addition to direct DNA damage by single- or double-strand breaks (SSB or DSB), the ionizing radiation causes indirect DNA damage by producing free radicals that reinforce the damage and make it permanent. However, the reduced production of free radicals under hypoxic conditions allows for the repair of the radiation-induced damage before it is “fixed” [3]. The abnormal blood vessels in the tumour also contribute to tumour chemoresistance by impeding drug perfusion and delivery. Additionally, in hypoxia, slow cell proliferation rates, as well as the upregulation of drug resistance-related genes, intensify the resistance of tumour cells [4].

For over a century there have been consistent efforts to target tumour hypoxia to promote radiation and chemotherapy responses. However, due to technical challenges and/or the lack of understanding of tumour biology, most trials failed in the clinical setting. Therefore, there is still a strong demand to solve these unmet needs to alleviate tumour hypoxia. In this review, we focus on the historical approaches to targeting hypoxia and how these can be better applied in the current times, in combination with new advances in techniques and molecular targeted treatments. A summary of the approaches discussed in this review is depicted in Figure 1.

## 2. Carbogen Breathing

The most relevant approach to overcome tumour hypoxia is to promote oxygen delivery. Hyperbaric oxygen (HBO) treatment consists of breathing 100% oxygen at 2–4 times the normal atmospheric pressure, which results in saturated haemoglobin and increased oxygen in the circulation [5,6,7]. While HBO is applied during or briefly before the radiation treatment, multiple clinical studies, including multi-centre randomised trials by the medical research council (MRC) in late 1970s and mid-1980s, showed that HBO treatment improved local control in patients with head and neck squamous cell carcinoma (HNSCC) or cervical cancer, compared to normal air. However, technical challenges derived from the decompression of the hyperbaric chamber, due to the radiation equipment and severe tissue damage, limited the use of HBO in radiation treatment [8]. Thus, carbogen breathing (95% oxygen + 5% carbon dioxide) was suggested as an alternative to HBO as it could promote oxygen delivery due to vasodilation and increased blood flow induced by carbon dioxide [9,10]. Nonetheless, carbogen breathing showed mixed results, which could be attributed to inconsistencies in the amounts of time patients were subjected to carbogen breathing [11,12]. Although carbogen treatment was well-tolerated by paediatric patients with high-risk brain stem glioma, no efficacy was found when combined with radiation [9].

Nicotinamide is a vitamin B3-derived molecule and its initial radiosensitising effect was thought to be mediated by the inhibition of the DNA damage repair (DDR) pathway specifically by blocking poly(ADP-ribose) polymerase 1 (PARP1), which is responsible for nucleotide excision repair (BER) [10]. The combination of carbogen with nicotinamide seems to target both acute and chronic hypoxia. Further studies suggested that nicotinamide could reduce acute hypoxia by inhibiting intermittent vascular shut down [11,12]. Accelerated radiotherapy with carbogen and nicotinamide (ARCON) trials showed a better patient survival and outcome in bladder and laryngeal cancer [13,14,15,16]. Although ARCON trials indicated the effectiveness of carbogen breathing as an adjuvant treatment regimen, it is necessary to consider other studies questioning the use of carbon dioxide in patients and the effectiveness of carbogen or the combination of carbogen and nicotinamide when compared to HBO [17,18].

## 3. Hyperthermia

Hyperthermia (HT) is a treatment based on applying heat (40 to 45°C) to tissues to above physiological temperatures. HT can be directly applied to treat tumours, but due to the technical difficulties in achieving cytotoxic temperatures, it is mainly used as an adjuvant therapy in combination with radiation or chemotherapy [19,20,21,22,23,24]. HT promotes protein denaturation and increases membrane fluidity and cell cycle inhibition. In addition to its direct action against tumour cells, HT is reported to inhibit DDR and promote tumour oxygenation, which might be more clinically relevant as they are directly related to tumour radiosensitivity [25,26].

Although HT has been used since ancient times to treat a variety of diseases. Its clinical use for cancer patients was introduced in the 1900s, which was accelerated by the development of the devices to heat tumours [27,28]. In patients, HT is externally or interstitially given using microwaves, radiofrequencies, electromagnetic radiation, or ultrasound. The accumulation of evidence supporting the beneficial effects from HT in cells, animals and patients eventually led to clinical trials in combination with radiation in the 1970s and 1980s. However, mixed patient outcomes put into question the efficacy of the combination of HT and radiation. Nevertheless, the analysis of five UK-led randomised trials of radiation and HT by Vernon et al., multi-centre trials by Overgaard et al., and studies by Jones et al. and van der Zee et al. suggested that the combination of HT and radiation was beneficial for some groups of patients, including locally recurrent breast cancer, cervical cancer, and melanoma [29,30,31,32,33,34,35,36,37,38,39]. Moreover, multiple studies further supported the beneficial effects of HT in paediatric and adolescent patients with a variety of tumour types, including malignant germ cell tumours, soft tissue sarcoma, and chondrosarcomas [40,41].

The biggest clinical challenge when delivering HT is to achieve cytotoxic temperatures (>43 °C) and to overcome thermotolerance [42,43]. It is hard to apply sufficient heat to induce cell death, without causing pain and damage to normal tissues. In addition, the isothermal (homogeneous) heating of tumour tissues is a major technical limitation in trying to reach the target temperature for the treatment. Non-homogeneous tissue heating also results in the differential heat resistance of tumour areas, which is caused by the formation of heat shock proteins (HSPs) [42]. Recent technical advances enabled real-time, MRI-guided thermotherapy (or MR thermometry) to define the heated region while monitoring the temperature rise [44,45]. The meta-analyses from breast cancer, cervical cancer, and HNSCC trials still support the advantage of HT with multimodal therapy [46,47,48]. Therefore, it is still necessary to fully understand the biological effects of HT and to set up a better strategy for the treatment schedule, patient groups, and the combination of other therapeutic modalities.

Multiple studies suggest that tumour oxygenation is caused by increased blood flow, perfusion, and decreased oxygen consumption, specifically at mild HT temperatures (39–43 °C) [30,49,50,51,52,53]. While tumours are often characterised by immature vascularisation, mild HT increases tumour blood flow and perfusion in order to increase oxygen delivery. A study using a rat tumour model determined that at 40.5–41.5 °C tumour oxygenation increased immediately and persisted even 24 h after treatment, which was correlated with changes in blood flow [54]. Interestingly, at 42.5 °C the trends of tumour oxygenation were different from those of blood flow. Although there was a decrease in blood flow 24 h after HT, tumour oxygenation was significantly enhanced at this temperature. This observation was supported by another study in spontaneous canine soft-tissue sarcomas [55]. These studies suggested that HT-induced tumour oxygenation was regulated by additional biological factors other than blood flow and perfusion. The study by Vaupel et al. revealed lower oxygen consumption rates at temperatures higher than 42°C [56]. After HT, dysregulated mitochondrial function and disrupted mitochondrial membrane potential were also observed [57,58,59]. In addition, the elevated lactate production in tumour cells or in mouse tissues after HT suggested that HT induced tumour glycolysis [24,60,61,62,63]. When combined, these studies indicate that HT alters cancer metabolism by inhibiting oxidative phosphorylation and promoting glycolysis, which results in a decreased oxygen consumption. The study by Moon et al. showed that mild HT lowered oxygen consumption, while the mitochondrial membrane potential was disrupted [60]. This study also indicated, using both in vitro and in vivo breast cancer models, that the high temperature-induced activation of HIF pathways at early time points (0–6 h after HT) was the main regulator of blood perfusion and metabolism. The radiosensitising effect of mild-HT-induced oxygenation was further observed in a fibrosarcoma model [61]. More interestingly, this study showed that having HT before radiation had a higher anti-tumour effect than having HT after radiation. This supports the idea that tumour oxygenation is required for radiosensitisation. While the previous clinical study by Jones et al. clearly supported the correlation between tumour oxygenation in mild HT temperatures and positive patient responses, recent clinical trials in advanced cervical cancer patients demonstrated that a short time interval between HT and radiation resulted in better patient outcomes [52,62]. This emphasises the importance of the treatment schedule.

The effect of HT in DDR is another aspect that can be exploited for radiosensitisation. HT was reported to induce increases in ATM and γH2AX pathways as well as p53 expression [23,63,64,65]. This induction in key pathways in DDR suggests a role in early DDR responses. The inhibition of non-homologous end joining (NHEJ) and homologous recombinant (HR) pathways by decreasing the expression of Ku80 and BRCA2 also suggests that HT might have a beneficial effect in combination with DDR-targeted therapy [37,66]. Currently, there is an ongoing clinical trial with the combination of HT and Olaparib in breast cancer patients with a chest wall recurrence (NCT03955640).

The modulation of the immune system by HT also suggests the potential of using HT as a systemic treatment. The increased augmentation of the major histocompatibility complex (MHC) class I, and the enhanced infiltration of NK cells, CD8+ T cells, and B cells were observed after HT [67,68,69,70]. These studies suggested that the combination of HT, radiation and immunotherapy may enhance treatment efficacy.

Taken together, mild HT has a strong potential as an adjuvant treatment to overcome tumour hypoxia. To obtain clearer insights into the relationship between HT treatment and tumour oxygenation at the molecular level, further investigation is required. Furthermore, the underlying mechanisms of heat resistance (heat shock proteins, HIF pathways, ROS, and endothelial-to-mesenchymal transition) need to be considered to identify therapeutic targets or to better stratify patient groups into benefitting from HT [60,71,72,73,74].

## 4. Hypoxia-Activated Prodrugs

In the 1980s, several studies proposed targeting hypoxia by developing hypoxia-specific cytotoxins, which are now known as hypoxia-activated prodrugs (HAPs) [75,76]. HAPs are compounds that can be selectively reduced by endogenous cellular oxidoreductases to yield cytotoxic agents under hypoxic conditions. This bioreductive process is normally inhibited by molecular oxygen, which directly competes for the single electron of the initially reduced prodrug, preventing the complete reduction of the compound into its active form. The superoxide product is then detoxified by superoxide dismutase (SOD) resulting in a low toxicity in normal tissues. However, under hypoxia, the prodrug is reduced by completely yielding the active compound, thus allowing for the specific target of hypoxic regions [77].

Since the 1960s, the discovery of a direct relation between the electron affinity of the sensitising agent and the degree of radiosensitisation led to the study of nitroaromatic compounds as potential hypoxia modifiers [78]. These types of HAPs present high electron affinities, which allow them to fix or stabilise radiation-induced DNA damage making it unrepairable, and thus lethal [79]. The chemical properties of nitroaromatic drugs under hypoxia resemble those of oxygen, hence, they have often been referred as “oxygen mimetics”.

The first nitroaromatic drugs, which were tested as radiosensitisers, were metronidazole and misonidazole. These nitroaromatic molecules showed a high effectiveness both in vitro and in preclinical models [80,81]. However, clinical trials failed for both compounds, showing no increases in patient survival between the combination treatment with radiation and radiation alone [82,83,84]. The failure of these trials was attributed to a high toxicity, which was caused by a high volume of distribution and a longer half-life of the drug in humans compared to murine models. As a result, the high toxicity of these compounds limited the maximal tolerable dose, which resulted in a low radiosensitisation. To increase the efficacy of the treatment, three new nitroaromatic compounds with improved pharmacokinetic properties and reduced toxicity were developed and tested clinically: etanidazole, pimonidazole, and nimorazole. Although etanidazole was involved in phase II and phase III clinical trials, it did not show any improvement for patients with neither HNSCC nor small-cell lung cancer (SCLC) [85,86]. Pimonidazole also failed to provide evidence of the benefit of a combination treatment with radiation, when compared to only radiation in cervical carcinoma patients [87]. However, currently, pimonidazole is widely used for measuring tumour hypoxia by immunohistochemical staining [88].

In contrast to etanidazole and pimonidazole, nimorazole showed promising clinical results. It improved the radiotherapeutic effect in supraglottic and pharynx tumours without side effects in a Danish head and neck cancer study (DAHANCA) [89]. As a result, nimorazole has now become a standard treatment for HNSCC in Denmark, although it is still necessary to assess whether nimorazole directly changes the oxygenation status of patient tumours. Recently, a phase I/II clinical trial for p16-negative HNSCC reported positive results for the combination of nimorazole and radiotherapy [90]. In addition to two ongoing clinical trials for nimorazole in Denmark (NCT02661152 and NCT02976051), there is a randomised clinical trial in the UK for patients with HNSCC undergoing radiotherapy who are not suitable for synchronous chemotherapy or cetuximab (NCT01950689).

## 5. Tirapazamine

Following the development of the initial set of oxygen-mimetic HAPs, new agents targeting both hypoxia and DNA were developed. Several compounds have been reported to reduce tumour hypoxia in vitro and were studied in the clinic, including mitomycin C, PR-104, evofosfamide (TH-302), A4QN, and tirapazamine (TPZ), as extensively reviewed elsewhere [91,92]. In this review, we specifically focus on TPZ as its usage to target tumour hypoxia has recently been revisited and is currently being clinically tested.

TPZ or 3-amino-1,2,4-benzotriazine1,4-dioxide is a HAP which can be reduced to its cytotoxic form by one electron reductases (i.e., NADPH-dependent cytochrome P450 reductase). This type of reduction leads to the formation of a TPZ radical and the release of hydroxide radical (OH·), then the TPZ radical is further reduced to TPZ 1-oxide by acquiring another electron [93,94]. TPZ can also be reduced by two electron reductases such as NAD(P)H dehydrogenase [quinone] 1 (NQO1) to TPZ 1-oxide directly [95]. The cytotoxicity of TPZ increases many fold under hypoxia. However, both forms of the reduced TPZ induce DNA damage through SSB and DSB, therefore TPZ can also be toxic under normoxia. The cytotoxic action of TPZ under normoxia is probably linked to the production of superoxides (O_2_^−^) and the activity of NQO1 [95]. NQO1 can reduce TPZ under normoxia, TPZ-1-oxide is then oxidized back to TPZ which produces O_2_^−.^; O_2_^−^ can then be converted into hydrogen peroxide or hydroxyl peroxide, which can damage normal tissues [96]. While the cytotoxic effect of TPZ is dependent on low oxygen tensions, tumour reperfusion (represented by tissue diffusion coefficients) and K values (the oxygen concentration is changed to halve the cytotoxic potency) also play a role in the prodrug activation and efficacy [97,98,99].

TPZ was found to have an additive cell killing effect with radiation and cisplatin tumour cell survival [100,101,102]. When given with cisplatin, which mainly targets normoxic tumour cells, TPZ increased therapeutic efficacy. Moreover, TPZ did not potentiate the cisplatin’s toxicity to normal tissues, which made it a potential candidate for clinical trials [103]. Phase I clinical trials of HNSCC, cervical cancer, and limited-stage SCLC showed promising results with the combination treatment of TPZ with cisplatin and radiation, as well as etoposide in the case of SCLC, decreasing the hypoxic tumour areas in patients [104,105,106,107]. Some trials showed the limited efficacy of TPZ in solid tumours. However, the majority of Phase I clinical trials suggested that TPZ, in combination with chemotherapy and radiation, compared to TPZ alone, had the potential to provide better tumour control [108,109]. The dose escalation studies for the TPZ showed cytotoxicity ranging from muscle cramping, vomiting and nausea, febrile neutropenia, and reversible deafness [108,109,110,111,112,113,114].

Phase II trials of TPZ also showed effectiveness when combined with cisplatin, radiation, or gemcitabine in HNSCC, non-small cell lung cancer (NSCLC), advanced or recurrent cervical cancer, and metastatic melanoma [110,111,112,113,114,115,116]. Additionally, the limited stage SCLC trial where TPZ was tested with etoposide, cisplatin and radiation, resulted in a better overall survival [117]. In contrast, a Phase II trial using radiation with TPZ on glioblastoma multiform did not show any differences in survival, indicating that there might be a tumour-type dependent action of TPZ [118].

Phase III clinical trials of cervical carcinoma did not find that TPZ and cisplatin to improve overall survival or progression free survival compared to cisplatin alone in stage IB2, IIA, IIB, IIIB, and IVA patients [119]. The advanced HNSCC trials also resulted in non-statistically significant differences between the combination therapy of cisplatin, TPZ and radiation and the combination of cisplatin and radiation [120]. A Phase III trial of the combination of cisplatin and TPZ in subjects with advanced, previously untreated, non-small cell lung tumours (CATAPULT I) showed a higher survival for the combination of TPZ and cisplatin compared to cisplatin alone. In contrast, the CATAPULT II trial showed that the combination of TPZ and cisplatin was less effective than the combination of etoposide and cisplatin [121,122]. Additionally, the CATAPULT II trial showed that the TPZ group also had higher toxicity. The results of Phase III trials were underwhelming and showed no benefit in having TPZ included in the treatment of the aforementioned cancer types. Furthermore, some even showed a higher toxicity when TPZ was included. However, it needs to be noted that none of these trials fully determined whether TPZ treatment promoted tumour oxygenation, which questions whether the drug was delivered properly and whether it can target hypoxic tumour cells effectively.

Interestingly, in paediatric cancers (refractory solid tumours and rhabdomyosarcoma), TPZ did not further increase toxicity when combined with cyclophosphamide or cyclophosphamide/doxorubicin [123,124]. However, in these studies, TPZ treatment failed to improve patient outcomes.

Although TPZ did not progress past the clinical trials due to the limited success in Phase III, recent studies suggest that TPZ might still be efficacious. Second generation TPZ analogues might protect cardiomycites from doxorubicin-induced toxicity [125]. Recently, it was found, in inflammatory breast cancer cell lines, that TPZ in combination with PARP1 inhibitors, olaparib and talazoparib, was synergistic [126]. Therefore, combining PARP1 inhibitors with TPZ could represent a new approach for targeting solid hypoxic tumours. Since TPZ-induced DNA damage was found to be repaired by enzymes involved in BER, it could be a logical next step to use PARP inhibitors and TPZ in combination [127]. Moreover, considering that the PARP inhibitor is also used in combination with HT, trimodality therapy with PARPi, TPZ, and HT might further enhance the radiosensitising effect.

Another possible usage of HAPs is to help deliver other anticancer agents. For instance, 2-nitroimidazole prodrugs were used to deliver the PARP1 inhibitor, 5-bromoisoquinolon [128]. Using multicellular layers (MCLs), a type of cell culture that models the extravascular compartment of tumours [1], a new generation of TPZ analogues have been developed. These analogues, such as SN30000, have higher tissue diffusion coefficients based on physicochemical properties such as molecular weight, lipophilicity, and hydrogen bond donors and acceptors [129]. Similar to the case of 2-nitroimidazole, it could be possible to deliver anticancer drugs and release them under hypoxia using new generations of HAPs. This could minimize chemotherapy-related toxicity while maximizing the direct targeting of tumour cells.

TPZ was also found to be capable of suppressing osteosarcoma cells under hypoxia in part through SLC7A11-mediated ferroptosis, which is a recently discovered type of cell death [130]. Interestingly, ferroptosis has been highlighted as a possible sensitiser of radioresistant cells [131]. Therefore, TPZ could be paired with ferroptosis inducers such as radiation and the ferroptosis inducer sulfasalazine [132]. Recently, preclinical studies of TPZ in treating liver tumours in transgenic mice, by inducing hypoxia through arterial embolization, have shown increased levels of tumour suppression [133]. The combination of TPZ and transarterial embolisation for unresectable hepatocellular carcinoma (HCC) patients is currently in clinical trials (NCT03145558) after the promising outcomes of a Phase I clinical trial [134].

## 6. Hypoxia-Inducible Factor (HIF) Inhibitors

At the molecular level, hypoxia stabilises the hypoxia-inducible transcription factor (HIF), which regulates more than 100 genes involved in tumour angiogenesis, apoptosis, invasion, metastasis and metabolism [135,136]. HIF consists of α and β subunits. Unlike the HIF-β subunit, which is constitutively active, the expression of the HIF-α subunit is regulated by oxygen, iron, free radicals and growth factors. Under normoxic conditions, HIF-α is hydroxylated by prolyl-4-hydroxylase (PHD) and bound to the von Hippel Lindau protein (VHL) to undergo proteasomal degradation by forming the E3 ligase complex [137,138,139]. Hypoxia (but also free radicals or metabolic intermediates) inhibits PHD activity, and thus stabilises HIF-α. When stabilised, HIF-α forms a heterodimer with β and binds to the hypoxia response element (HRE) with cofactors CBP and p300 to transactivate the expression of many downstream genes [140].

HIF pathways are highly upregulated in most tumour types and their expression is correlated with poor patient outcomes [135,141,142,143]. The expression of HIF-α is also upregulated after radiation or chemotherapy, indicating that HIF pathways are significantly involved in treatment resistance [144,145,146,147,148]. Therefore, there is a strong rationale to target HIF pathways to sensitise hypoxic tumour cells to treatment.

Multiple attempts have been made to target HIF to inhibit hypoxia-driven cancer progression. However, only those indirectly acting on HIF are currently used in clinical trials, including camptothecin and bortezomib [149,150]. Other small molecules in clinical trials such as the Hsp90 inhibitor (17-AAG), digoxin, and 2-methoxyestradiol (2ME2) also act non-specifically on HIF through the inhibition of HIF synthesis or stabilisation [151,152,153,154,155]. On the contrary, PT2385 and its derivative, PT2399, specifically bind to the per-arnt-sim-B (PAS-B) domain of HIF2 and interfere in its heterodimersation with HIF-β [156]. The efficacy of PT2399 was proven in a preclinical clear-cell renal cancer cell (ccRCC) model, which exhibited an enhanced HIF activation due to VHL mutation [157,158]. The recent outcome of a Phase I clinical trial showed both the safety and efficacy of PT2385 as a direct inhibitor of HIF-2 in ccRCC. In contrast, in vivo studies using a glioblastoma model showed no benefits in survival when comparing PT2385 treatment combined with radiation (RT)/temozolomide (TMZ) to RT/TMZ treatment [159]. Despite this, PT2977 (MK-6482), a more potent second-generation HIF-2 inhibitor, is currently in a Phase III clinical trial in advanced RCC (NCT04195750) [160,161,162,163]. It is important to note that the tumour oxygenation status was not evaluated in any of these studies, indicating that further investigation is required to determine whether this treatment is more beneficial to the specific patient population with a high HIF-2 expression or elevated tumour hypoxia. The ongoing Phase II clinical trial (NCT03216499) with glioblastoma patients will provide further information on whether PT2385 can be used to target HIF and hypoxia to enhance patient treatment outcome.

## 7. Targeting Metabolism

An emerging alternative approach to alleviate tumour hypoxia in order to increase radiosensitivity is to reprogramme the cellular metabolism into lower oxygen consumption rates (OCR). Oxygen is mostly used by mitochondria through oxidative phosphorylation (OXPHOS) and the electron transport chain (ETC). Therefore, reducing cellular OCR by inhibiting OXPHOS in well-perfused and peri-hypoxic areas of tumours increases oxygen availability and facilitates the diffusion of the non-utilised oxygen into hypoxic areas, reducing tumour hypoxia and increasing radiation responses [164,165].

There are several advantages to reducing tumour hypoxia by lowering OCR in tumours. Computational analysis indicated that inhibiting oxygen consumption inadequately perfused tumour regions, with a decrease as little as 30%, was more effective at reducing tumour hypoxia than increasing blood flow or elevating the oxygen concentration in blood [166]. Another key advantage is that reducing oxygen consumption does not require the diffusion of drugs into hypoxic and poorly vascularised areas, as OXPHOS inhibitors act on well-perfused areas allowing oxygen to infiltrate and oxygenate hypoxic areas. Additionally, this strategy can be broadly applied to different types of hypoxic tumours, as inhibiting OXPHOS reduces OCR in several types of cancer [164].

Various treatments that aim to target the ETC to impair OXPHOS and decrease oxygen consumption are currently being tested. One of these treatments is the commonly prescribed, anti-diabetic biguanide, metformin. Metformin was first highlighted as a potential cancer treatment after the observation that diabetic patients treated with metformin had a lower cancer incidence and better outcome than patients with or without diabetes taking other medications [167,168,169,170]. Two different mechanisms of lowering tumour growth were attributed to metformin: the reduction in insulin availability at the organismal level and the inhibition of complex I of the ETC [170]. Complex I inhibition decreases ATP production activating AMP-activated protein kinase (AMPK) and inhibiting the mechanistic target of the rapamycin complex 1 (mTORC1), hence, reducing the overall OCR [171,172]. In fact, metformin was shown to decrease OCR, tumour hypoxia and increase radiosensitisation in vitro, in spheroids and in xenografted tumours [173,174,175,176,177]. Moreover, there are two ongoing phase II clinical trials for metformin as a tumour oxygenating agent in locally advanced cervical cancer (LACC) (NCT04275713 and NCT02394652) and a proof-of-principle clinical trial for HNSCC (NCT03510390). Additionally, metformin is also currently being tested in a phase I clinical trial for paediatric relapsed or refractory solid tumours (NCT01528046).

One of the controversies around using metformin to reduce tumour hypoxia is that the OCR reduction reported was considerably low (<20%). Hence, a more potent biguanide, phenformin, was proposed as an alternative. Phenformin was reported to decrease OCR levels and reduce radioresistance in a xenograft model of colorectal cancer [178]. Similarly, phenformin was shown to reduce tumour growth through the complex I inhibition in mouse models from a variety of cancer types [175,179,180,181,182,183]. A Phase I clinical trial is currently recruiting melanoma patients to test the efficacy of phenformin in combination with BRAF and MEK inhibitors (NCT03026517). Similarly, several new compounds are reported to target complex I with a high affinity and specificity, including BAY87-2243 [184], ME-344 [185] and IACS-010759 [186,187]. These complex I-inhibitors were introduced in phase I clinical trials. However, the clinical trial for BAY87-2243 was terminated due to toxicity (NCT01297530) and the trial for ME-344 resulted in no clinical efficacy [188]. IACS-010759 is currently under phase I clinical trials in solid tumours (NCT03291938) and relapsed/refractory acute myeloid leukaemia (AML) (NCT02882321). Interestingly, IACS-010759 was tested in combination with radiotherapy and anti-PD-1 in PD-1-resistant NSCLC, resulting in a prolonged survival time [189]. However, it is yet unclear whether the prolonged survival effect is directly related to the alleviation of tumour hypoxia.

Arsenic trioxide is a well-known inhibitor of the complex IV in the ETC; it is an FDA-approved treatment for acute promyelocytic leukaemia (APL) and is involved in numerous clinical trials in other cancer types. It was shown that arsenic trioxide reduced tumour hypoxia in Lewis lung carcinoma (LLC) and transplantable mouse liver (TLT) with a consequent increase in radiotherapy response [190]. Arsenic trioxide affects numerous intracellular signal transduction pathways and alters several cellular functions [191]. Therefore, it is still unclear whether this drug is capable of reducing OCR levels to alleviate tumour hypoxia at the concentration used in clinic.

Atovaquone is an FDA-approved anti-malarial agent that also showed OCR inhibition and its subsequent tumour hypoxia reduction in vitro and in vivo. This ubiquinone analogue acts as a complex III inhibitor reducing cellular oxygen consumption without inducing cell death. Atovaquone was reported to reduce the OCR by more than 80% in several types of cancer in vitro, as well as a significant tumour growth reduction in xenograft models [173]. The first proof-of-principle clinical trial with atovaquone as a tumour hypoxia treatment for NSCLC (NCT02628080) was finalised with highly promising results, indicating that atovaquone can reduce tumour hypoxia and exert antitumoral effects at the mRNA level [192]. Hypoxia imaging using PET-CT and dynamic contrast-enhanced (DCE) MRI clearly showed a reduced hypoxic tumour volume and increased perfusion in atovaquone-treated patients. Atovaquone is currently undergoing a Phase I clinical trial in combination with chemoradiation in NSCLC (NCT04648033). In addition, its possible use in paediatric cancer patients is suggested by a study showing the safety of atovaquone as a preventive treatment for *Pneumocistis carinii* pneumonia in children with leukaemia [193]. Other anti-parasitic drugs such as ivermectin, mefloquine, quinacrine, and proguanil were tested in vitro and in vivo as mitochondrial function inhibitors and showed a potential to decrease tumour hypoxia [194].

Overall, targeting OXPHOS to reduce tumour hypoxia and increase radiosensitivity represents a shift away from focusing on hypoxic regions and overcomes the need to supply drugs into diffusion-limited hypoxic areas. Although at an early stage, several ETC inhibitors have been developed and are being tested clinically with promising results and will hopefully lead to new therapies to alleviate tumour hypoxia.

## 8. Conclusions

For over a century, extensive efforts have been made to target tumour hypoxia in both experimental and clinical settings. It is critical to develop treatments that aim to reduce low oxygen environments within tumours and inhibit tumour progression and treatment resistance. To overcome hypoxia, multiple attempts were made to increase oxygen delivery, decreasetumour oxygen consumption, specifically target hypoxic cells, and inhibit HIF pathways. Although most of the trials did not accomplish a successful translation into the clinic, these studies still support the importance of hypoxia targeting to promote tumour control and patient survival. Technological advances and a better comprehension of tumour biology revived some of these historical treatments. Here, we reviewed several examples of these advances, such as the development of real time MRI-guided thermotherapy (MR thermometry). MR thermometry allows for the monitoring and control of tumour tissue temperature during HT, hence improving the application of HT. We also reviewed several second-generation molecules with improved pharmacokinetics and a higher potency, such as nimorazole or the HIF-2 inhibitor, PT2977, which showed promising results. Furthermore, we discussed the alternative usage of treatments that failed in clinical trials but are now being tested in combination with other inhibitors, such as TPZ in combination with PARP1 inhibitors. Lastly, we highlighted examples of drugs that could be repurposed, such as the anti-diabetic metformin and the anti-malarial atovaquone. These compounds were shown to reduce tumour hypoxia and are currently being tested as radiosensitisers. Therefore, we hope that combining efforts to find old and new remedies to target hypoxia will soon translate into more effective therapeutic regimens.

## Figures and Tables

**Figure 1 biomolecules-11-01604-f001:**
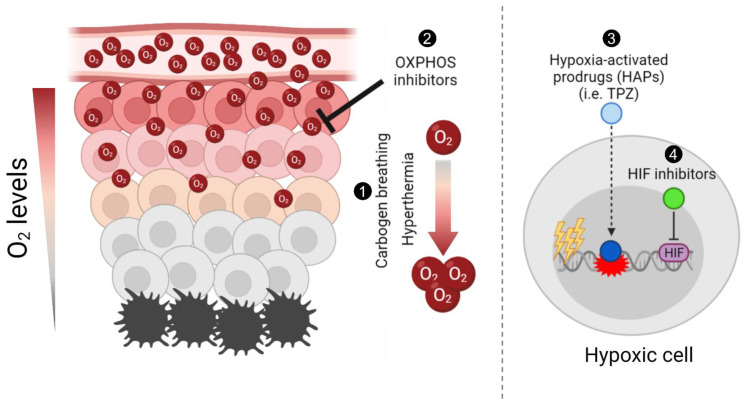
Schematic summary of different approaches to target tumour hypoxia. Briefly, (1) carbogen breathing and hyperthermia aim to increase the overall oxygen concentration in tumours, enhancing radiosensitisation; (2) Oxidative phosphorylation (OXPHOS) inhibitors target mitochondrial metabolism in well-oxygenated cells to promote an increase and diffusion of the overall oxygen into hypoxic regions and enhance radiation-induced damage; (3) hypoxia-activated prodrugs (HAPs), such as tirapazamine (TPZ), are reduced into their active cytotoxic form in hypoxic cells; and (4) the hypoxia-inducible factor (HIF) inhibitors aim to inhibit the hypoxia-induced activation of HIF, hence selectively targeting hypoxic cells. The figure was generated using Biorender.com (Accessed on 21 October 2021).

## Data Availability

Not applicable.

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
