# Peer review of "Targeting Hypoxia: Revival of Old Remedies"

_biomolecules, 2021, doi:10.3390/biom11111604_

Round 1
Reviewer 1 Report
The authors have provided a short review on the potential benefit of resurrecting older methods for modulating hypoxia for treatment of solid tumors. While these approaches may not have the general interest of new approaches, they certainly warrant attention. The summaries generally informative and balanced. There are, however, some issues that need to addressed.
- line 78- this sentence seems to be missing something. Are tehy raising tumor temperature?
- line 82- is membrane fluidity increased?
- line 85 says that HT has been used since ancient times. For what?
- line 91- what are UK-leading trials?
- line 95-96- While heat shock response is mentioned later, it could be useful to discuss it briefly here also.
- line 107 - 109- leaky vessels are not a real counterpoint to increased flow and oxygen delivery.
- line 115- reoxygenation is not really the right term. This has specific meaning that does not really apply to a tumor that never had normal oxygenation.
- line 135- as written, this is not really a sentence. Should the hyphen in HT-induced be deleted.
- line 274- should be stabilizes rather than activates.
- line 319-322- Oxygen does not readily diffuse through aquesous media. That is why capillary density is so important.
- Overall, the paper is fairly well organized but it would benefit by careful editing for English usage.
Author Response
Reviewer 1
The authors have provided a short review on the potential benefit of resurrecting older methods for modulating hypoxia for treatment of solid tumors. While these approaches may not have the general interest of new approaches, they certainly warrant attention. The summaries generally informative and balanced. There are, however, some issues that need to be addressed.
: Thank you so much for your review and comments. We are pleased to address to your concerns and comments as shown below.
- line 78- this sentence seems to be missing something. Are they raising tumor temperature?
: Thank you for the comment and we apologise that it was not clear what information we tried to deliver. Now we change the sentence as “Hyperthermia (HT) is a treatment based on applying heat (40 to 45°C) to tissues to above physiological temperatures”
- line 82- is membrane fluidity increased?
: Thank you for your comment. To clarify the meaning of the sentence now it is modified as “HT promotes protein denaturation, increases in membrane fluidity,~”
- line 85 says that HT has been used since ancient times. For what?
: I apologise if the sentence was not clear. Now the sentence is modified as “HT has been used since ancient times to treat a variety of diseases”
- line 91- what are UK-leading trials?
: Thank you for the comment. To clearly address the sentence, now it is modified to “Nevertheless, the analysis of five UK-lead randomised trials of radiation and HT by Vernon et al.,~”
- line 95-96- While heat shock response is mentioned later, it could be useful to discuss it briefly here also.
: We appreciate reviewer’s great suggestion. Now the sentence is modified to “The biggest clinical challenge when delivering HT is to achieve cytotoxic temperatures (>43°C) and to overcome thermotolerance [43,44]” and we also added a new sentence “Non-homogeneous tissue heating also results in differential heat resistance of tumour areas, which is caused by the formation of heat shock proteins (HSPs) [43]” in line 102 with new references (Li et al. 1995. Int J Hyperthermia, Overgarrd & Nielson. 1983. Radiother Oncol)
- line 107 - 109- leaky vessels are not a real counterpoint to increased flow and oxygen delivery.
: Thank you for the comment. We removed a word “leak” and now the sentence is modified as “While tumours are often characterised by immature vascularisation~”
- line 115- reoxygenation is not really the right term. This has specific meaning that does not really apply to a tumor that never had normal oxygenation.
: We apologise that we did not clearly deliver the information using the correct term. Now the phrase is modified as “tumour oxygenation”.
- line 135- as written, this is not really a sentence. Should the hyphen in HT-induced be deleted.
: We apologise the typo that we made. Now the sentence is modified to “HT was reported to induced increases in ATM ~”
- line 274- should be stabilizes rather than activates.
: We apologise that we did not correctly phrase the sentence. We modified sentence as “hypoxia stabilises the hypoxia-inducible factor (HIF)~”
- line 319-322- Oxygen does not readily diffuse through aqueous media. That is why capillary density is so important.
: Thank you for the correction. Now we modified the sentence as “OXPHOS inhibitors act on well-perfused areas allowing for oxygen to infiltrate and oxygenated hypoxic areas”
- Overall, the paper is fairly well organized but it would benefit by careful editing for English usage.
: Again, thank you so much for reviewer’s constructive and positive comments.
Reviewer 2 Report
Authors clearly presented published works reporting the efforts to discover effective treatments for tumour hypoxia. Authors focused on old remedies and their promising application in the oncological setting.
The manuscript is well written and presented. However, manuscript may be improved. Please, take into account the following suggestions.
- Authors stated that technological advances and a better comprehension of tumour biology have revived some of the historical treatment, but it is not clear to me from the manuscript what kind of evidences revived some of the historical treatment supporting this conclusion. Comparisons with more recent solutions may provide more adequate support to your conclusion.
- Authors should mention how and whether clinical studies have assessed the presence\absence of tumour hypoxia or not. This is an important point that should be reported and discussed.
- Authors mentioned that “For over a century there have been consistent efforts to target tumour hypoxia to promote radiation and chemotherapy responses. However, several issues are still unresolved due to technical challenges and/or the lack of understanding of tumour biology”. This sentence seems to contradict in some way authors’ conclusions. Please, clarify this point.
- Could you please report, if any, some successful or promising application of old remedies to in paediatric cancers?
Author Response
Reviewer 2
Authors clearly presented published works reporting the efforts to discover effective treatments for tumour hypoxia. Authors focused on old remedies and their promising application in the oncological setting.
The manuscript is well written and presented. However, manuscript may be improved. Please, take into account the following suggestions.
: Thank you so much for reviewer’s review and comments. We addressed to reviewer’s suggestions below.
- Authors stated that technological advances and a better comprehension of tumour biology have revived some of the historical treatment, but it is not clear to me from the manuscript what kind of evidence revived some of the historical treatment supporting this conclusion. Comparisons with more recent solutions may provide more adequate support to your conclusion.
: Thank you for the suggestion. Now in conclusion, we added summary of current technological and biological advances to repurpose historical treatment in combination with new hypoxia targeted therapies.
- Authors should mention how and whether clinical studies have assessed the presence\absence of tumour hypoxia or not. This is an important point that should be reported and discussed.
: We appreciate reviewer’s comments about hypoxia status of patients in clinical trials with hypoxia-targeted treatment. We also agree that it is important information that needs to be addressed to evaluate whether treatments were delivered to clinically reduce hypoxia. In the main text, we highlighted clinical studies that reported patient tumour oxygenation (hyperthermia, tirapazamine, atovaquone) and those that did not provide information about tumour hypoxia.
- Authors mentioned that “For over a century there have been consistent efforts to target tumour hypoxia to promote radiation and chemotherapy responses. However, several issues are still unresolved due to technical challenges and/or the lack of understanding of tumour biology”. This sentence seems to contradict in some way authors’ conclusions. Please, clarify this point.
: We apologise if we did not clearly deliver the sentence. Now, the sentence is modified as “However, due to technical challenges and/or the lack of understanding of tumour biology, most trials failed in clinical setting. Therefore, there is still a strong demand to solve these unmet needs to alleviate tumour hypoxia.”
- Could you please report, if any, some successful or promising application of old remedies to in paediatric cancers?
: Thank you for great suggestion, since toxicities of treatment in paediatric cancers will be the important factor to be considered. Now we added sentences describing several clinical trials using hypoxia targeting agents or treatment to treat paediatric cancers. Newly added sentences are highlighted in the text.
Round 2
Reviewer 2 Report
Authors have adressed all issues